# Numerical approaches for the rapid analysis of prophylactic efficacy against HIV with arbitrary drug-dosing schemes

**Lanxin Zhang**, **Junyu Wang**, **Max von Kleist***

Project group 5 "Systems Medicine of Infectious Disease", Robert Koch Institute, Berlin, Germany

* kleistm@rki.de

## Abstract

Pre-exposure prophylaxis (PrEP) is an important pillar to prevent HIV transmission. Because of experimental and clinical shortcomings, mathematical models that integrate pharmacological, viral- and host factors are frequently used to quantify clinical efficacy of PrEP. Stochastic simulations of these models provides sample statistics from which the clinical efficacy is approximated. However, many stochastic simulations are needed to reduce the associated sampling error. To remedy the shortcomings of stochastic simulation, we developed a numerical method that allows predicting the efficacy of arbitrary prophylactic regimen directly from a viral dynamics model, without sampling. We apply the method to various hypothetical dolutegravir (DTG) prophylaxis scenarios. The approach is verified against state-of-the-art stochastic simulation. While the method is more accurate than stochastic simulation, it is superior in terms of computational performance. For example, a continuous 6-month prophylactic profile is computed within a few seconds on a laptop computer. The method's computational performance, therefore, substantially expands the horizon of feasible analysis in the context of PrEP, and possibly other applications.

## Author summary

Pre-exposure prophylaxis (PrEP) is an important tool to prevent HIV transmission. However, experimental identification of parameters that determine prophylactic efficacy is extremely difficult. Clues about these parameters could prove essential for the design of next-generation PrEP compounds. Integrative mathematical models can fill this void: Based on stochastic simulation, a sample statistic can be generated, from which the prophylactic efficacy is estimated. However, for this sample statistic to be accurate, many simulations need to be performed.

Here, we introduce a numerical method to directly compute the prophylactic efficacy from a viral dynamics model, without the need for sampling. Based on several examples with dolutegravir (DTG) -based short- and long-term PrEP, as well as post-exposure prophylaxis we demonstrate the correctness of the new method and its outstanding computational performance. Due to the method's computational performance, a number of

**Data Availability Statement:** All code written in support of this publication is publicly available at https://github.com/KleistLab/PrEP.git.

**Funding:** L.Z. and M.v.K. acknowledge funding from the German Ministry for Science and Education (BMBF; grant number 01KI2016). The funders had no role in study design, data collection and analysis, decision to publish, or preparation of the manuscript.

**Competing interests:** The authors have declared that no competing interests exist.

analyses, including formal sensitivity analysis, are becoming feasible with the proposed method.

This is a *PLOS Computational Biology* Methods paper.

## Introduction

Since its transfer to human in the early 20th century [1], HIV remains a major public health threat. According to UNAIDS estimates, approximately 38 million individuals worldwide are infected with the human immunodeficiency virus (HIV) [2]. HIV continues to spread and the latest incidence estimates amount to about 1.7 million new infections in 2019 [2]. Sub-Saharan Africa is hit hardest by the HIV pandemic, and due to COVID many services, including HIV control and treatment, had been suspended, which could lead to a long-term re-surge in infections [3].

Nowadays, about 30 antiviral compounds are available that can stop HIV replication and prevent the acquired immunodeficiency symdrome (AIDS) and AIDS-related death [4]. However, unlike many other infections, no cure is available to clear HIV, which can persist in latent reservoirs for decades [5, 6]. Available treatments therefore have to be taken life-long to prevent the relapse of virus from latent reservoirs and to prevent AIDS. As a consequence, much focus around fighting HIV turned towards HIV prevention. While a vaccine would be the ideal tool for the purpose, intrinsic difficulties have so far precluded the development of an effective vaccine against HIV [7]. However, based on the successes in antiviral drug discovery, recent years have seen an increasing interest in utilising antivirals not only for treatment, but also to prevent HIV transmission. Two general strategies are currently implemented for this purpose: (i) Treatment-as-prevention (TasP) intends to put individuals with an HIV diagnosis immediately on treatment, which essentially makes them non-contagious by decreasing the number of viruses they can expose to uninfected individuals [8]. (ii) Pre-exposure prophylaxis (PrEP) on the other hand prevents establishment of HIV infection after exposure [9, 10].

Currently, two oral PrEP options, the patent-expired two-compound combination Truvada, as well as the patent-protected two-compound combination Descovy are available. However, many more drugs are investigated for re-purposing [11, 12], or under *de novo* development [13], including topically-applied drugs, long-acting injectibles, as well as drug eluting implants [14, 15].

However, demonstrating clinical efficacy of novel PrEP compounds constitutes a formidable task. Clinical efficacy of PrEP is understood as the reduction in the number of infected individuals in a treatment- vs. a control arm of a clinical trial [9]. A major statistical problem arises from the fact that HIV transmission probabilities are extremely low (e.g. $< 3\%$ during unprotected sex [16]; far less for condom- or PrEP usage, and when potential transmitters take antiviral therapy). Hence, the number of evaluable data points (= infected individuals in a trial treatment and control arm) are extremely low and prone to chance events. Since the approval of Truvada-based PrEP, novel PrEP interventions have to be compared with Truvada, worsening the statistical problem considerably [17]. E.g. the recent DISCOVER trial evaluating the efficacy of emtricitabine plus tenofovir alafenamide (Descovy) against Truvada was conducted over 8756 person-years [10], yielding as little as 22 evaluable data points (infections).

Statistically empowering such a study quickly exceeds organizational and monetary capacities. The statistical limitation has two consequences: (a) The determination of concentration-prophylaxis relations, threshold concentrations and contributions of transmitted and acquired drug-resistance cannot be rigorously deduced from clinical data, non-withstanding ethical concerns. (b) The hurdles to introduce next-generation PrEP regimen are immense: Trials consume huge monetary resources and require several years. This compromises the advancement of next-generation PrEP and likely affects its costs. It is therefore absolutely crucial to discern promising from less promising interventions *a priori*.

Auxiliary tools based on integrative mathematical modelling may help to better understand the parameters contributing to clinical PrEP efficacy [18]. In particular, how drug dosing may alter the risk of acquiring HIV infection, depending on its timing and the magnitude of viral exposure.

A key feature of HIV biology is that transmission is highly inefficient. For example, $< 3\%$ of unprotected sex acts between sero-discordant partners result in HIV infection [16]. Moreover, the number of genetically distinct founder viruses is extremely low [19]. This argues that stochastic processes play an important role during early infection, and that, in the majority of exposures, the virus becomes eliminated before it irreversibly infects the new host. Therefore, stochastic modelling and simulation approaches are used to estimate the efficacy of PrEP by integrating various host-, viral- and drug-specific parameters. For fixed drug concentrations, Monte-Carlo schemes, analytical, as well as probability generating ODE systems have been developed [20–22]. These approaches have been extended to include time-varying drug concentrations by integrating pharmacokinetic characteristics, as well as realistic dosing schemes, but were restricted to particular drugs, drug classes or prophylactic schemes [23, 24]. Recently, a numerically exact Monte-Carlo approach was introduced that can be universally applied to study the effects of dosing, pharmacokinetics, drug adherence, timing and extend of viral exposure on the risk of HIV infection [11, 12]. Despite its advantage, the introduced stochastic simulation approach is still computationally prohibitive, in a sense that it would not allow to compute a PrEP efficacy from a history of drug dosing 'on the fly', e.g. to be useful in a health app or computer program that empowers PrEP users, akin to [25].

In this work, we derive a numerical method from an established viral dynamics model of HIV infection [26, 27] that overcomes aforementioned limitations. The method estimates the probability of viral extinction using a set of low-dimensional deterministic Ansatz functions that are solved with standard numerical solvers. The method allows to quantify PrEP efficacy within fractions of a second on a standard computer and is numerically exact up to the tolerance level of an ODE-solver. This method fully integrates drug pharmacokinetics, which allows to estimate the influence of drug dosing, drug adherence, timing and magnitude of viral exposure on PrEP efficacy.

We illustrate the method with the second-generation integrase inhibitor dolutegravir (DTG). Taking advantage of the outstanding performance of the developed method, we presented several show cases to display their possible applications: By estimating (i) both pre- and post-exposure prophylaxis with different dosing of the drug, as well as timing- and extent of virus exposure, (ii) prophylactic protection profiles over a 6 month dosing history, as well as (iii) timing- and probability of viral establishment with different exposures.

## Methods

The initial replication of HIV after exposure is highly stochastic. Typically, a low number of founder viruses is responsible for establishing infection and the transmission probability per *sexual* exposure is very low. In biology two types of stochastic noise are considered, roughly

categorized as internal- and external noise. Herein we focused on the internal noise, which assumes that the stochastic outcome of viral exposure can be explained by the order in which the considered reactions occur. For example, when a virus comes in close proximity of target cells, it can be either eliminated or infect a target cell and replicate. Prophylactic drugs shift the balance of these two events towards viral clearance. Stochastic dynamics of this type can be defined by a multivariate Poisson process, and the evolution of the state probabilities can be then described by the chemical master equation (CME). Conditioned on an initial state $x_0$, the CME can be written down for any particular state $x_i$ as:

$$\frac{d}{dt}\mathbb{P}(X_t = x_i \mid X_0 = x_0) = \sum_{k=1}^{K} a_k(x_i - \nu_k) \cdot \mathbb{P}(X_t = x_i - \nu_k \mid X_0 = x_0)$$
$$- a_k(x_i) \cdot \mathbb{P}(X_t = x_i \mid X_0 = x_0)$$

with $t \geq 0$ and $\nu_k$ denoting the stoichiometry of reactions yielding state $x_i$. Essentially, the CME as stated above, locally describes the flux of probability into- and out of the state $x_i$, with propensity functions $a_k$. $X_t \in \mathbb{N}^s$ denotes the state of the system (the combination of numbers of viral compartments, as well as drug particles), where $s$ denotes the overall number of variables. Because each variable can take any value in the natural numbers (e.g. 0, 1, 2, . . ., $\infty$), the CME denotes an infinite dimensional system of ordinary differential equations, that is intractable (*curse of dimensionality*).

In the context of estimating PrEP efficacy, we are however not interested in the entire state space but only interested in estimating the probability of extinction of all viral compartments, as outlined below. Thus, the key idea of our proposed method is to derive a low-dimensional system of ordinary differential equations, that allows to estimate only those probabilities relevant to estimating PrEP efficacy.

## Prophylactic efficacy

The prophylactic efficacy $\varphi$ is defined as the reduction of infection risk for a prophylactic regimen $\mathcal{S}$, compared to the infection probability in the absence of prophylaxis:

$$\varphi(Y_t, \mathcal{S}) = 1 - \frac{P_I(Y_t, \mathcal{S})}{P_I(Y_t, \varnothing)} \tag{1}$$

where $P_I(Y_t, \mathcal{S})$ and $P_I(Y_t, \varnothing)$ denote the infection probabilities in the presence and absence of a prophylactic regimen $\mathcal{S}$ for a given virus state (e.g. exposure) $Y_t$ at time $t$. We consider a prophylactic regimen to be a continuous function of drug concentrations. The state of the viral compartments $Y_t$ is defined as $Y_t = [V, T_1, T_2]^T$, where $V$, $T_1$ and $T_2$ are the numbers of viruses, early stage infected T cells $T_1$ and late infected T cells $T_2$, respectively. For the absence of prophylaxis, $P_I(Y_t, \varnothing)$, analytical solutions have been presented in [28]. For a prophylactic regimen $\mathcal{S}$ they need to be determined numerically.

The infection probability is the complement of the extinction probability $P_E$. We thus have

$$P_I(Y_t, \mathcal{S}) = 1 - P_E(Y_t, \mathcal{S}) \tag{2}$$

where

$$P_E(Y_t) := \mathbb{P}\left( Y_\infty = \begin{bmatrix} 0 \\ 0 \\ 0 \end{bmatrix} \,\middle|\, Y_t = \begin{bmatrix} V \\ T_1 \\ T_2 \end{bmatrix} \right).$$

In words, the probability that all viral compartments will eventually go extinct, starting from state $Y_t$ at time $t$. Under the reasonable assumption of statistical independence, akin to

[28], we can define the extinction probability as:

$$P_E(Y_t, \mathcal{S}) = (P_E(Y_t = \hat{V}, \mathcal{S}))^V \cdot (P_E(Y_t = \hat{T}_1, \mathcal{S}))^{T_1} \cdot (P_E(Y_t = \hat{T}_2, \mathcal{S}))^{T_2} \qquad (3)$$

where $\hat{V}, \hat{T}_1, \hat{T}_2$ represent the unit vectors:

$$\hat{V} = \begin{bmatrix} 1 \\ 0 \\ 0 \end{bmatrix}, \ \ \hat{T}_1 = \begin{bmatrix} 0 \\ 1 \\ 0 \end{bmatrix}, \ \ \hat{T}_2 = \begin{bmatrix} 0 \\ 0 \\ 1 \end{bmatrix} \qquad (4)$$

Thus, if $P_E(Y_t = \hat{V}, \mathcal{S}), P_E(Y_t = \hat{T}_1, \mathcal{S})$ and $P_E(Y_t = \hat{T}_2, \mathcal{S})$ can be determined, the prophylactic efficacy can also be calculated using Eqs (1)–(3).

Below, we will present a method to compute the extinction probability numerically. First we will introduce a within-host HIV dynamics model, as well as a pharmacokinetic-pharmacodynamic model of the second-generation integrase dolutegravir, which serve as a common basis to derive and demonstrate the presented numerical method to compute HIV prophylactic efficacy. We will then introduce the proposed method (the Probability Generating System; PGS). A formal derivation of the method can be found in S1 Text. A pseudo-code for the method is found in S2 Text. To highlight the generality of the method, we derive the equations for the PGS for a different viral dynamics model that involves latently infected cell dynamics in S3 Text.

## HIV viral dynamics model

We adapted the viral dynamics model from [26, 27]. We use this model, because it allows to mechanistically integrate the mechanisms of action of all approved drugs (and drug classes) [26] and because it allows to integrate both drug-specific *in vitro* and *in vivo* parameters [24]. In its most basic form, the considered viral replication cycle consists of free infectious viruses V, early infected T-cells ($T_1$), and productively infected T-cells ($T_2$). The dynamics can be defined by six reactions $R_1$ to $R_6$ with propensities $a_1$-$a_6$:

$$R_1 : \text{Clearance of free virus}, \quad V \rightarrow * \quad a_1 = (CL + CL_T \cdot T_u) \cdot V \qquad (5)$$

$$R_2 : \text{Clearance of } T_1\text{-cell}, \quad T_1 \rightarrow * \quad a_2 = (\delta_{PIC} + \delta_{T_1}) \cdot T_1 \qquad (6)$$

$$R_3 : \text{Clearance of } T_2\text{-cell}, \quad T_2 \rightarrow * \quad a_3 = \delta_{T_2} \cdot T_2 \qquad (7)$$

$$R_4 : \text{Infection of a suscept. cell}, \quad V \rightarrow T_1 \quad a_4 = \beta \cdot T_u \cdot V \qquad (8)$$

$$R_5 : \text{Integration of viral DNA}, \quad T_1 \rightarrow T_2 \quad a_5(\varnothing) = k \cdot T_1 \qquad (9)$$

$$R_6 : \text{Production of new virus}, \quad T_2 \rightarrow V + T_2 \quad a_6 = N_T \cdot T_2, \qquad (10)$$

where we assume that the integrase inhibitor dolutegravir (DTG) inhibits proviral genome integration (reaction $R_5$), with details outlined below. Moreover, we assume that a $T_2$-cell continuously produces viruses (with reaction rate $R_6$) until it is cleared (continuous virus production model). The basic model is depicted in Fig 1. Utilized model parameters and their interpretations are given in Table 1.

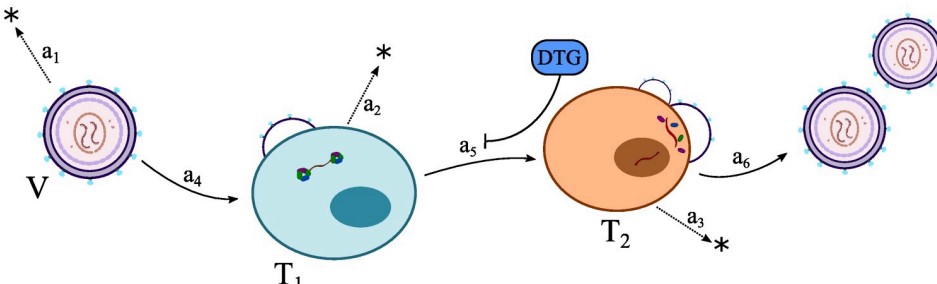

**Fig 1. Schematic of the utilized viral dynamics model.** V, $T_1$, $T_2$ denote virus, early infected T-cells and productively infected T-cells respectively. Each reaction is denoted by its reaction propensity $a_1 - a_6$. Briefly, a free virus can either be cleared (with reaction propensity $a_1$), or infect a susceptible T cell with rate $a_4$ to yield an early infected cell $T_1$. These cells denote a state where the virus has penetrated the host cell, but has not yet integrated its proviral DNA into the host cell's genome, thus not yet producing viral offspring. Early infected cells $T_1$ can either be cleared with rate $a_2$ or the proviral DNA irreversibly becomes integrated into the host cells DNA with rate $a_5$ to yield a productively infected T-cell $T_2$. $T_2$ cells start producing infectious progeny virus with rate $a_6$, or they may get cleared by the immune system with rate $a_3$.

## Pharmacokinetics and pharmacodynamics of dolutegravir

Dolutegravir (DTG) is a second-generation integrase inhibitor which may potentially be used as prophylaxis against HIV. Moreover, we study it because of its similarity to carbotegravir, which is in clinical development as a next-generation PrEP compound. To evaluate the PrEP utility of DTG, we utilize a previously developed pharmacokinetic-pharmacodynamic model of the drug [11].

**Pharmacokinetics of dolutegravir.** We utilize the non-linear mixed effects pharmacokinetic model introduced in [11]. In brief, a two-compartment model with first order absorption describes the plasma concentrations time profiles of dolutegravir (DTG) after oral drug administration. Individual parameters for a population of HIV-negative individuals were sampled from the distributions defined in [11] (Table 2 therein). The structural pharmacokinetic

**Table 1. Parameters for viral dynamics model.**

| Parameter | Description | Value | Reference |
|---|---|---|---|
| $CL$ | clearance rate of free virus by the immune system | 2.3 | [29, 30] |
| $CL_T$ | clearance rate of the free virus during unsuccessful infection | $CL_T = \left(\frac{1}{\rho_{rev}} - 1\right) \cdot \beta$ | [26] |
| $T_u$ | level of uninfected T-cells in the disease-free state | $T_u = \lambda_T / \delta_T$ | |
| $\rho_{rev}$ | probability of successful reverse transcription | 0.5 | [31, 32] |
| $\beta$ | lumped rate of infection of T-cells | $8 \cdot 10^{-12}$ | [33] |
| $\lambda_T$ | birth rate of uninfected T-cells | $2 \cdot 10^9$ | [34] |
| $\delta_T$ | death rate of uninfected T-cells | 0.02 | [35] |
| $\delta_{PIC}$ | rate of intracellular destruction of pre-integration complex (PIC); | 0.35 | [32, 36] |
| $\delta_{T_1}$ | rate of clearance of $T_1$-cells | 0.02 | [35] |
| $\delta_{T_2}$ | rate of clearance of $T_2$-cells | 1 | [37] |
| $k$ | rate by which $T_1$-cells are transformed into $T_2$-cells | 0.35 | [32] |
| $N_T$ | rate of production of infectious progeny virus | 670 | [26, 35] |

All parameters are in units [1/day], except for $\lambda$ [cells/day] and $\beta$ [1/(day · virus)]

model is given by the following set of ordinary differential equations (ODEs):

$$\frac{d}{dt}Z_1 = -k_a \cdot Z_1 \tag{11}$$

$$\frac{d}{dt}D = \frac{d}{dt}Z_2 = \frac{k_a}{V_c/F_{bio}} \cdot Z_1 - \frac{CL/F_{bio}}{V_c/F_{bio}} \cdot Z_2 - \frac{Q/F_{bio}}{V_c/F_{bio}} \cdot Z_2 + \frac{Q/F_{bio}}{V_p/F_{bio}} \cdot Z_3 \tag{12}$$

$$\frac{d}{dt}Z_3 = \frac{Q/F_{bio}}{V_c/F_{bio}} \cdot Z_2 - \frac{Q/F_{bio}}{V_p/F_{bio}} \cdot Z_3 \tag{13}$$

where $Z_1$ represents the amount of drug in the dosing compartment, and $Z_3$ denotes the DTG concentration in the peripheral compartment. $D = Z_2$ is the DTG concentration in the blood plasma, i.e. the value of interest. In a therapy, the value $Z_1$ increases whenever a dosing event $\tau_k$ occurs: $Z_{1,t} = Z_{1,t} + \text{dose}_k$.

**Pharmacodynamics of DTG.**  Since DTG is an integrase inhibitor, it acts intracellularly by preventing the integration of proviral DNA. This effect can be translated into a decrease in propensity function $a_5$ by a factor $1 - \eta_D$

$$a_5(t) = (1 - \eta_D(t)) \cdot k \cdot T_1 \tag{14}$$

where $\eta_D(t)$ denotes the direct effect of DTG at time $t$, which is modelled using the Emax-equation [38]:

$$\eta_D(t) = \frac{D_t^m}{IC_{50}^m + D_t^m} \tag{15}$$

where $D_t$ is the drug concentration in the blood plasma at time t. $IC_{50}$ represents the plasma drug concentration by which the activity of proviral integration is inhibited by 50%, and $m$ denotes a hill coefficient. In this work $IC_{50} = 89$ [nM] and $m = 1.3$ is used, which are values after protein adjustment [28] (free drug hypothesis).

## Low-dimensional deterministic Ansatz function

In this section, we present a method to compute the extinction probability for the unit vectors $P_E(Y_t = \hat{V}, \mathcal{S})$, $P_E(Y_t = \hat{T}_1, \mathcal{S})$ and $P_E(Y_t = \hat{T}_2, \mathcal{S})$, which enable the integration of arbitrary pharmacokinetic profiles resulting from some prophylactic regimen $\mathcal{S}$. As noted before, this would enable calculating prophylactic efficacy for arbitrary drug/dosing regimen.

**Distribution of state transition events.**  The viral dynamic model illustrated in Fig 1 is interpreted as a continuous time, discrete state Markov process [39]. Therefore, the time when a particular state transition happens is exponentially distributed according to the reaction propensities. In the viral replication cycle, if the initial state is $Y_0 = [1, 0, 0]^T$, there are two possible next states: (i) the virus is cleared $Y_\tau = [0, 0, 0]^T$, or a $T_1$-cell emerges $Y_\tau = [0, 1, 0]^T$, where $\tau$ is a random, exponentially distributed waiting time, until a reaction fires. The probability density function (PDF) for state transition $V \rightarrow T_1$ can be derived as:

$$\begin{aligned} f_{V \rightarrow T_1}(x) &= (1 - F_{a_1}(x)) \cdot f_{a_4}(x) \\ &= (1 - (1 - e^{-a_1 x})) \cdot (a_4 e^{-a_4 x}) \\ &= a_4 e^{-(a_1 + a_4)x} \end{aligned} \tag{16}$$

where $F_{a_1}(x)$ is the cumulative probability of state transition $V \rightarrow \varnothing$ between time point 0 and

$x$, and $f_{a_4}(x)$ is the probability density function for transition $V \rightarrow T_1$. In words: The probability that $V \rightarrow T_1$ occurs and $V \rightarrow \varnothing$ has not occurred yet. Corresponding derivations hold for the process $T_2 \rightarrow T_2 + V$:

$$
\begin{aligned}
f_{T_2 \rightarrow T_2 + V}(x) &= (1 - F_{a_3}(x)) \cdot f_{a_6}(x) \\
&= (1 - (1 - e^{-a_3 x})) \cdot (a_6 e^{-a_6 x}) \\
&= a_6 e^{-(a_3 + a_6)x}
\end{aligned}
\tag{17}
$$

For the process $T_1 \rightarrow T_2$, the probability distribution $f_{a_5}(x)$ is different since the values of $a_5$ are time-dependent when an integrase inhibitor is applied (as in our example) that affects the reaction according to its pharmacokinetic-pharmacodynamic (PK-PD) properties and its dosing history. Using a Taylor approximation, the probability distribution for $f_{T_1 \rightarrow T_2}(x)$ can be derived as (S1 Text):

$$
f_{T_1 \rightarrow T_2}(x) = (1 - F_{a_2}(x)) \cdot f_{a_5}(x) = a_5(x) e^{-(a_2 x + \int_0^x a_5(t)\ dt)}
\tag{18}
$$

**Probability Generating System (PGS).** The precondition for the PGS method is that the concentration profile of the prophylactic regimen $\mathcal{S}$ must be known in advance. This can be achieved by solving the deterministic pharmacokinetic Eqs (11)–(13) for a particular dosing schedule using standard ODE-solvers. Given these pharmacokinetic profiles, the PGS method delivers an extinction probability-time profile. Mathematically, the method calculates $P_E(Y_t, \mathcal{S})$.

First, we derive the discrete form of this method. The discrete form is based on a time-discretization of the underlying time-continuous Markov process into time steps $\Delta t$. From the constructed discrete-time Markov chain, we can compute the extinction probabilities for e.g. state $\hat{V}$ at time point $t$ as follows:

$$
\begin{aligned}
P_E(Y_t = \hat{V}) = \ & P(Y_{t+\Delta t} = \mathbf{0} \mid Y_t = \hat{V}) + \\
& P(Y_{t+\Delta t} = \hat{T}_1 \mid Y_t = \hat{V}) \cdot P_E(Y_{t+\Delta t} = \hat{T}_1) + \\
& P(Y_{t+\Delta t} = \hat{V} \mid Y_t = \hat{V}) \cdot P_E(Y_{t+\Delta t} = \hat{V})
\end{aligned}
\tag{19}
$$

with the following interpretations: $P(Y_{t+\Delta t} = \mathbf{0} \mid Y_t = \hat{V})$ denotes the probability that the virus is eliminated in time span $[t, t + \Delta t]$; $P(Y_{t+\Delta t} = \hat{T}_1 \mid Y_t = \hat{V})$ is the probability that the state transition $V \rightarrow T_1$ occurs in $[t, t + \Delta t)$ and $P_E(Y_{t+\Delta t} = \hat{T}_1)$ is the probability that a $T_1$-cell that exists at time $t + \Delta t$ will eventually be eliminated. Finally, the probability that no state transition occurs in $[t, t + \Delta t)$ is given by

$$
P(Y_{t+\Delta t} = \hat{V} \mid Y_t = \hat{V}) = 1 - P(Y_{t+\Delta t} = \mathbf{0} \mid Y_t = \hat{V}) - P(Y_{t+\Delta t} = \hat{T}_1 \mid Y_t = \hat{V}).
$$

The terms $P(Y_{t+\Delta t} = \mathbf{0} \mid Y_t = \hat{V})$ and $P(Y_{t+\Delta t} = \hat{T}_1 \mid Y_t = \hat{V})$ can be derived based on the calculations for the distribution of state transition events, as outlined in S1 Text. Using these

derivations, the the final expression for $P_E(Y_t = \hat{V})$ is

$$
\begin{aligned}
P_E(Y_t = \hat{V}) = \quad &\frac{a_1}{a_1 + a_4}\left(1 - e^{-(a_1+a_4)\Delta t}\right) \\
&+ \frac{a_4}{a_1 + a_4}\left(1 - e^{-(a_1+a_4)\Delta t}\right) \cdot P_E\left(Y_{t+\Delta t} = \hat{T}_1\right) \\
&+ e^{-(a_1+a_4)\Delta t} \cdot P_E\left(Y_{t+\Delta t} = \hat{V}\right)
\end{aligned}
\tag{20}
$$

Similarly, for $P_E(Y_t = \hat{T}_1)$ we get:

$$
\begin{aligned}
P_E(Y_t = \hat{T}_1) = \quad &\frac{a_2}{a_2 + a_5(t)}\left(1 - e^{-(a_2+a_5(t))\Delta t}\right) \\
&+ \frac{a_5(t)}{a_2 + a_5(t)}\left(1 - e^{-(a_2+a_5(t))\Delta t}\right) \cdot P_E\left(Y_{t+\Delta t} = \hat{T}_2\right) \\
&+ e^{-(a_2+a_5(t))\Delta t} \cdot P_E\left(Y_{t+\Delta t} = \hat{T}_1\right)
\end{aligned}
\tag{21}
$$

and for $P_E(Y_t = \hat{T}_2)$:

$$
\begin{aligned}
P_E(Y_t = \hat{T}_2) = \quad &\frac{a_3}{a_3 + a_6}\left(1 - e^{-(a_3+a_6)\Delta t}\right) \\
&+ \frac{a_6}{a_3 + a_6}\left(1 - e^{-(a_3+a_6)\Delta t}\right) \cdot P_E\left(Y_{t+\Delta t} = \hat{T}_2\right) \\
&\cdot P_E\left(Y_{t+\Delta t} = \hat{V}\right) + e^{-(a_3+a_6)\Delta t} \cdot P_E\left(Y_{t+\Delta t} = \hat{T}_2\right)
\end{aligned}
\tag{22}
$$

Having Eqs (20)–(22), we will take $\Delta t \to 0$ to derive a set of continuous Ansatz functions. With this idea in mind, we differentiate Eqs (20)–(22) to derive the following set of ordinary differential equations (S1 Text):

$$
\begin{aligned}
\frac{dP_E(Y_t = \hat{V})}{dt} = \ &a_1 \cdot (P_E(Y_t = \hat{V}) - 1) \\
&+ a_4 \cdot (P_E(Y_t = \hat{V}) - P_E(Y_t = \hat{T}_1)) \\
\frac{dP_E(Y_t = \hat{T}_1)}{dt} = \ &a_2 \cdot (P_E(Y_t = \hat{T}_1) - 1) \\
&+ a_5(t) \cdot (P_E(Y_t = \hat{T}_1) - P_E(Y_t = \hat{T}_2)) \\
\frac{dP_E(Y_t = \hat{T}_2)}{dt} = \ &a_3 \cdot (P_E(Y_t = \hat{T}_2) - 1) + a_6 \cdot (P_E(Y_t = \hat{T}_2) \\
&- P_E(Y_t = \hat{T}_2) \cdot P_E(Y_t = \hat{V}))
\end{aligned}
\tag{23}
$$

Given $a_5(t)$ and initial values, equations above can be solved by any ODE solver, as outlined in S2 Text.

## Implementation and availability

Pseudocodes for the method are described in detail in S2 Text. The method was implemented in Python 3.8, using SciPy 1.5.0, Numpy 1.18.5, Pandas 1.0.5 and matplotlib 3.2.2. Codes are available from https://github.com/KleistLab/PrEP.git.

## Algorithmic specifications

**Probability Generating System (PGS).** The simulation time horizon must exceed the time horizon of interest. In this work, we extended the end time point for an extra $\tau = 100$ hours so that the values in the target time interval are accurate. This value of extra time can be determined roughly from the half life of considered drugs, e.g. $\tau = 7 \cdot t_{1/2}$, where $t_{1/2} = 14.5$ hours denotes the half life of DTG. The above stated criteria guarantees that the drug concentrations at $T_e + \tau$ are $< 1\%$ of the trough concentrations. The method is solved backwards from an end time $T_e + \tau$ to some start time $T_s$. At $T_e + \tau$, we initialize the ODE with the values in the absence of drugs, e.g. $P_E(Y_{T_e} = \hat{V}, \mathcal{S}) = P_E(\hat{V}, \varnothing)$, $P_E(Y_{T_e} = \hat{T}_1, \mathcal{S}) = P_E(\hat{T}_1, \varnothing)$ and $P_E(Y_{T_e} = \hat{T}_2, \mathcal{S}) = P_E(\hat{T}_2, \varnothing)$, which can be determined analytically [21]. We use `solve_ivp` in SciPy [40] with the LSODA solver (linear multistep method) and default settings to solve the system of ODEs *backwards* in time.

**EXTRANDE.** We implemented the exact stochastic simulation method EXTRANDE with configurations identical to [11].

**Pharmacokinetics.** Pharmacokinetic profiles for DTG were pre-computed from Eqs (11)–(13) using `solve_ivp` in SciPy [40] with default solver and default settings, for the respective prophylactic regimens $\mathcal{S}$ and $D_t$ was linearly interpolated. Using the pharmacokinetic profiles, the time-dependent value of $\eta_D(t)$, Eq (15) was determined and used in the respective algorithms. Depending on the analysis, we either simulated DTG pharmacokinetics for a representative individual, or by drawing pharmacokinetic parameters for 1000 virtual individuals from the parameter distributions defined in [11].

## Simulation of pre- and post-exposure prophylaxis

**Single viral challenge.** For single viral challenges the profiles of $P_E(Y_t = \hat{V}, \mathcal{S})$, $P_E(Y_t = \hat{T}_1, \mathcal{S})$ and $P_E(Y_t = \hat{T}_2, \mathcal{S})$ were computed using PGS (Eq (23)) with configurations outlined above. To compute extinction probabilities with arbitrary viral challenges we used Eq (3) and computed prophylactic efficacy from there using Eqs (1) and (2).

**Multiple viral challenges.** Using the new methods, we can also compute the prophylactic efficacy following multiple viral challenges: Under the assumption of statistical independence [28], the extinction probability following multiple viral challenges is the product of extinction probabilities for single viral challenges at their corresponding time points. I.e.:,

$$\varphi(Y_{\{t_i\}}, \mathcal{S}) = 1 - \frac{P_I(Y_{\{t_i\}}, \mathcal{S})}{P_I(Y_{\{t_i\}}, \varnothing)}. \tag{24}$$

with $\{t_i\}$, $i = 1, \ldots, n$ denotes a set of $n$ viral exposures and

$$P_I(Y_{\{t_i\}}) = 1 - \prod_i P_E(Y_{t_i})$$

due to statistical independence, where $P_E(Y_{t_i})$ is computed as described above.

If $n$ viral exposures occur at the same time $t_i = t$, we have

$$P_I(Y_{\{t_i\}}) = 1 - (P_E(Y_t))^n.$$

To assess the validity of this assumption we also simulated multiple viral challenges with EXTRANDE and compared the results.

### Computation of density function of the extinction event

So far, we computed the probability that viral extinction is eventually happening after viral exposure at some time $t$; $P_E(Y_t = \hat{V}, \mathcal{S})$. I.e., we only care if extinction eventually occurs, regardless of *when* it happens.

The introduced method can however also be used to estimate the probability that the extinction occurs in a specific time range. To solve for this probability, we can e.g. alter the initial conditions of the PGS. In essence, we are interested in the extinction probability within a time range $[t_0, t_e]$. Since Eq (23) are solved backwards in time (S2 Text), we initialize the set of ODEs with the probability that the extinction occurs at the final time point $t_e$, for example $P(Y_{t_e} = \mathbf{0} \mid Y_{t_e} = \hat{V})$ for the first ODE (Eq (23)). This probability is obviously 0. Consequently, the set of ODEs Eq (23) is initialized with values [0, 0, 0] and then solved backwards in time until $t_0$. The derived probabilities have the following interpretation: they denote the probability that a single virus, single $T_1$ and single $T_2$, transmitted at time point $t_0$ are cleared in the time range $[t_0, t_e]$. For example, if we want to determine the probability that virus is cleared $t_e = 10$ days after exposure to one infectious virus (at day $t_0$) for some prophylactic regimen, we set the time span of the ODE set Eq (23) to [0, 10], initialize the ODE with $[0, 0, 0]^T$ at $t_e = 10$ and solve the PGS backwards to $t_0 = 0$ to derive $P(Y_{t_e} = \mathbf{0}, \mathcal{S} \mid Y_{t_0} = \hat{V})$ in the first equation of the PGS, Eq (23). For a given pharmacokinetic (PK) profile $D_t$ this process has to be repeated for different values of $t_e$ to reconstruct the entire cumulative probability density function (CDF) with the correct PK profile. The probability density function (PDF) can be derived straightforward from the CDF. Similarly, the procedure also allows to compute the corresponding probabilities for arbitrary numbers of initial viruses (statistical independence assumption), Eq (3).

## Results

### Model predictions in the absence of drugs

To assess the validity of the proposed method and the utilized model, we initially performed some simulations without any drugs.

Using the PGS, we calculated the probability of viral extinction when one virus $P_E(Y_t = \hat{V}, \varnothing)$, one early infected cell $P_E(Y_t = \hat{T}_1, \varnothing)$ cell and one late infected cell $P_E(Y_t = \hat{T}_2, \varnothing)$ were inoculated respectively.

The respective extinction probabilities were 90.17%, 52.07% and 1.50%, respectively, which is identical to the analytical solutions (constant drug concentrations including drug absence) presented in [21].

We then computed the density function of the extinction event, as shown in Fig 2A using different inoculum sizes. Essentially, when a single virus reaches a replication-competent compartment (red line in Fig 2A), if extinction occurs, it occurs very early after exposure, with the probability declining exponentially. When 10 viruses are inoculated (green line in Fig 2A), viral extinction is most likely to occur 14 hours after exposure. Overall extinction is less likely

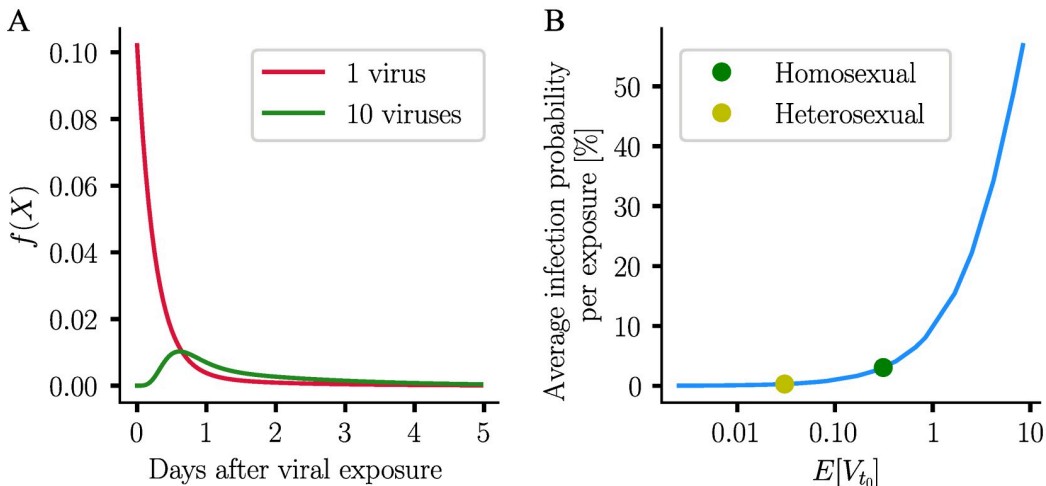

**Fig 2. Model simulations in the absence of drugs. A**. Probability density function of the extinction event in absence of drug, for one initial virus and 10 initial viruses respectively. **B**. Relation between the expected inoculum size and the corresponding average infection probability per exposure, in the absence of drugs. The distribution of inoculum sizes is described in [24]. Average infection probabilities for homosexual vs. heterosexual transmission [16], are marked by dots.

to occur with 10 viruses than with an inoculum of 1 virus. With both inocula, if viral extinction happens, it has to happen before day 5 post-exposure.

Finally, we wanted to estimate the average infection probabilities $\mathbb{E}_{Y_t}[P_I(Y_t, \varnothing)] = 1 - \mathbb{E}_{Y_t}[P_E(Y_t, \varnothing)]$ for different inoculum sizes and compare with published data, Fig 2B. To that end, we used viral load distributions $P(\mathrm{VL} = k)$ in potential transmitters, combined with the virus exposure model from [24] to computed *average* infection probabilities in case of heterosexual vs. homosexual intercourse.

$$\mathbb{E}_{Y_t}[P_E(Y_t, \varnothing)] = \sum_{n=0}^{\infty} P_E\left(Y_t = \begin{bmatrix} n \\ 0 \\ 0 \end{bmatrix}, \varnothing\right) \underbrace{\int_{k=0}^{\infty} P(\mathrm{VL} = k) \cdot P(V_0 = n | \mathrm{VL} = k)}_{\text{probability to inoculate } n \text{ viruses}}$$

The simulations with the PGS in the absence of drugs (Fig 2B) indicate a clear increase of the infection risks with increasing virus exposure. Moreover, the predictions capture reported average per-exposure infection risks of 3% (homosexual intercourse) vs. 0.3% (heterosexual intercourse) [16] quite accurately.

Next, we evaluated predictions, where we utilized actual pharmacokinetics of DTG to compute the prophylactic efficacy with the PGS method for different dosing regimen.

## Efficacy of PrEP-on-demand with DTG

Using the proposed method, we computed the time course of the extinction probability $P_E(Y_t, \mathcal{S})$ and the corresponding prophylactic efficacy $\varphi(Y_t, \mathcal{S})$ for a 3-days once daily short-course oral 50mg DTG prophylaxis, that was either initiated shortly after viral exposure (post-exposure prophylaxis, PEP) or before virus exposure (pre-exposure prophylaxis, PrEP). Fig 3A shows the profiles of the extinction probability and Fig 3B depicts the corresponding prophylactic efficacy. In Fig 3, for illustration, we depict these quantities for $Y_t = \hat{V}$, $Y_t = \hat{T}_1$ and

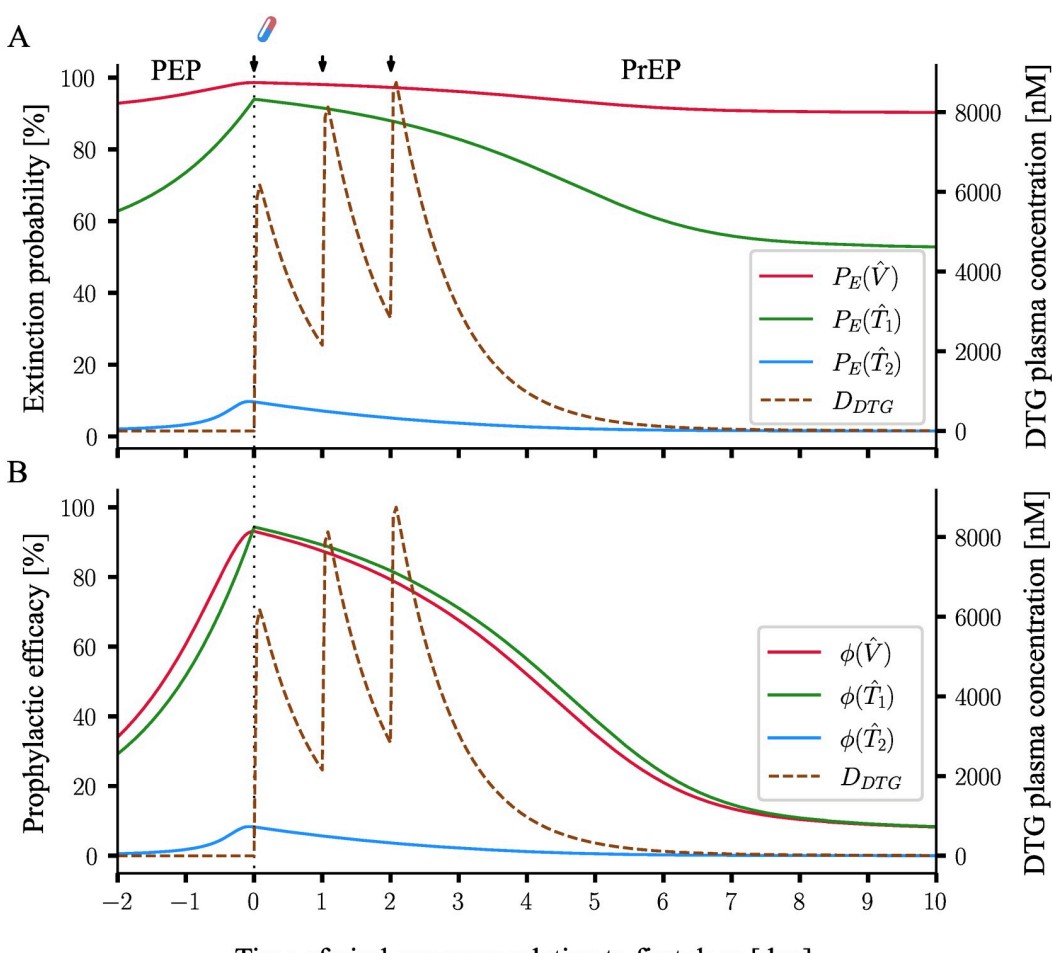

**Fig 3. Prophylactic efficacy for 3-days once daily short-course oral 50mg DTG.** The extinction probabilities were computed by PGS, for a representative individual with pharmacokinetic parameters $k_a$ = 2.24h$^{-1}$, $V_p/F_{bio}$ = 0.73 L, $Q/F_{bio}$ = 0.0082 L/h, $CL/F_{bio}$ = 0.85 L/h and $V_c/F_{bio}$ = 17.7 L. Observation began from two days before the first dose (the drug-doses are marked by arrows), until the 10th day after the first dose of DTG. The X-axis denotes the timing of viral exposure relative to the first dose, i.e. negative values represent a viral exposure before the first dose of DTG (post-exposure prophylaxis, PEP), whereas positive values represent pre-exposure prophylaxis (PrEP) scenarios. **A**. DTG plasma concentration (dashed brown line) and the extinction probability profiles, with regards to one virus $P_E(\hat{V}, \mathcal{S})$, one $T_1$-cell $P_E(\hat{T}_1, \mathcal{S})$ and one $T_2$-cell $P_E(\hat{T}_2, \mathcal{S})$ are shown by solid red, green and blue lines. **B**. DTG plasma concentration (dashed brown line) and the corresponding prophylactic efficacies for one virus $\varphi(\hat{V}, \mathcal{S})$, one $T_1$-cell $\varphi(\hat{T}_1, \mathcal{S})$ and one $T_2$-cell $\varphi(\hat{T}_2, \mathcal{S})$ are shown by solid red, green and blue lines.

$Y_t = \hat{T}_2$ individually. From these quantities, the extinction probability for an arbitrary initial state can be calculated based on Eq (3).

From a computational point of view, PEP and PrEP are computed within the same execution of the proposed methods: A pharmacokinetic trajectory (brown dashed line in Fig 3A and 3B) is placed on the time axis and the methods are backwards propagated to some time before the first dose of the drug was given. Any time points *before* the first dose denote PEP, whereas PrEP refers to time points *after* the first dose of the drug. For example, the value of the red line in Fig 3B at $t = -2$ (days) indicates that the efficacy of a 3 days 50mg oral DTG post-exposure prophylaxis, that was initiated 2 days after exposure to a single virus particle, is about 35%. If exposure occurred at $t = 0$ (coinciding with the first drug intake), the efficacy would be $\approx$ 90%.

As described above, the new methods therefore compute a prophylactic efficacy profile for a time span of interest, i.e. every point on this time-efficacy curve represents the prophylactic efficacy $\varphi(Y_t, \mathcal{S}_{t_0})$ conditioned that the viral exposure occurred at the indicated time point and the prophylactic regimen $\mathcal{S}$ was started at $t_0 = 0$.

From a biological point of view, Fig 3B nicely highlights the role of the molecular target of the integrase inhibitor DTG within the viral replication cycle [28]: The drug is able to potently prevent infection if it emanates from a virus $V$ or a $T_1$ cell (compartments *preceding* its molecular target process), but not from a $T_2$ (a compartment *succeeding* its molecular target process).

On a AMD R5 core with 3.6 Ghz and 16 GB RAM, PGS ran in a fraction of a second for the presented example in Fig 3. Notably, if stochastic simulation was performed (as in [11]), several thousand stochastic samples need to be generated to approximate the prophylactic efficacy $\varphi(Y_t, \mathcal{S})$ from the sample statistic for a single time point $t$.

## Comparison with stochastic simulation (EXTRANDE)

Previously, an exact hybrid stochastic simulation method called EXTRANDE was introduced in [41]. We subsequently adapted the algorithm to estimate PrEP efficacy against HIV [11, 12]. Here, we used EXTRANDE to verify the accuracy of the proposed method. Fig 4 shows the predicted efficacy of a three days prophylaxis with either 2 or 50mg oral DTG started at $t = 0$ using EXTRANDE vs. PGS. In contrast to Fig 3 (drug-centered evaluation), here we perform an exposure-centered evaluation to calculate $\varphi(Y_{t_0} = \hat{V}, \mathcal{S}_{t_i})$. I.e. the virus exposure occurs

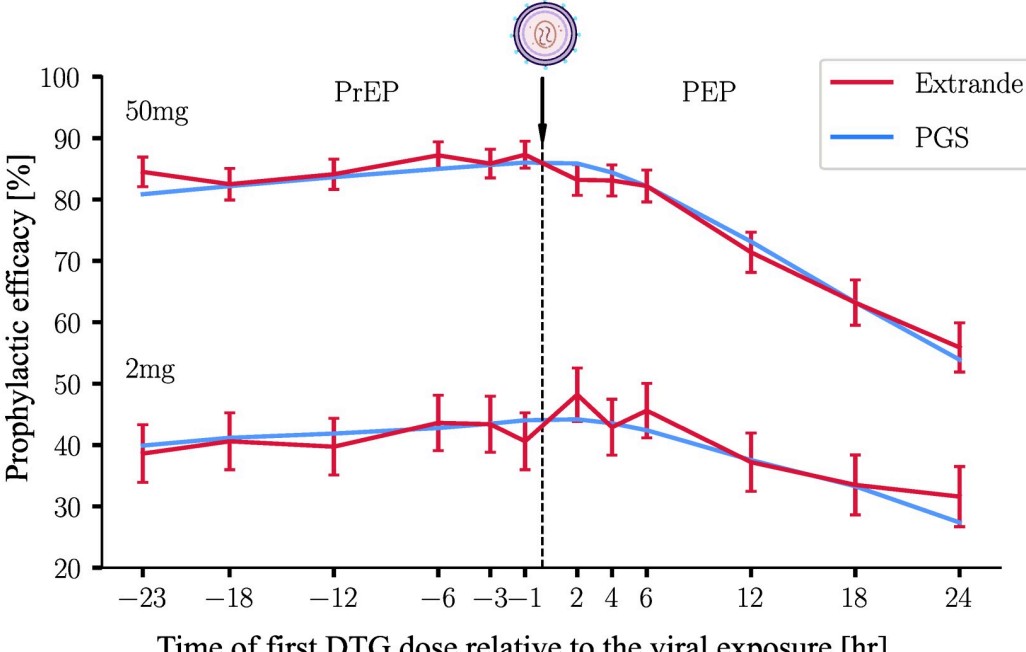

**Fig 4. Prophylactic efficacy for PrEP and PEP computed by EXTRANDE and the new methods.** 2 mg (lines below) and 50 mg (lines above) DTG were ingested for three days, respectively. The extinction probabilities were computed by EXTRANDE and PGS, for a representative individual with pharmacokinetic parameters $k_a = 2.24h^{-1}$, $V_p/F_{bio} = 0.73$ L, $Q/F_{bio} = 0.0082$ L/h, $CL/F_{bio} = 0.85$ L/h and $V_c/F_{bio} = 17.7$ L. The prophylactic efficacy was computed using Eq (1) for initial state $Y_0 = [1, 0, 0]^T$ (exposure to a single virus particle). The X-axis represents the timing of the first DTG dose relative to the virus challenge, which is marked by the arrow. EXTRANDE was run 10 000 times for each condition. The error bars denote the 95% confidence bounds for the ensemble estimate, computed using the Greenwood's formula [42].

$t_0 = 0$ and the first dose is taken at time $t_i \in \{-23, -18, -12, -6, -3, -1, 2, 4, 6, 12, 18, 24\}$ hours before/past the viral exposure. In words: the prophylactic efficacy if exposure to a single virus particle occurred at time $t_0 = 0$ and the 3-day prophylaxis was initiated at time $t_i$.

In Fig 4, we can see that the results of PGS form smooth lines, whereas the results of EXTRANDE fluctuate randomly around the results of PGS. From a biological standpoint we see that the prophylactic efficacy 50mg DTG is almost double compared to 2mg. Moreover, we see that the prophylactic efficacy deteriorates much faster for PEP than for PrEP. I.e., if DTG is taken as PEP, it needs to be taken shortly after the exposure. For PrEP-on demand, the efficacy changes only marginally if PrEP is initiated within 24 hours prior to exposure.

In terms of run time, EXTRANDE requires thousands of simulations to achieve statistically reliable meaningful results. In Fig 4, for each of the 12 time points we ran 10 000 simulations with EXTRANDE, which took about two hours for all points using multi-treading (12 treads). By contrast, using the proposed method, the values of all time points can be extracted from a single run, i.e. $\varphi(Y_{t_0} = \hat{V}, \mathcal{S}_{t_i}) = \varphi(Y_{-t_i} = \hat{V}, \mathcal{S}_{t_0})$, with run times less than one second.

Obviously, the pharmacokinetic profiles can be arbitrarily altered, which allows to assess the prophylactic efficacy of any regimen $\mathcal{S}$, e.g. with regards to the drugs taken, their dose, the administration frequency and the timing of drug intake, as outlined above.

## PrEP efficacy for multiple viral challenges and different inoculum sizes

Another interesting application of the proposed method is to assess the impact of the exposure on the prophylactic efficacy, e.g. to assess the sensitivity of $\varphi(Y_t, \mathcal{S}_{t_0})$ with regards to $Y_t$. We have shown previously by simulations [23] that the prophylactic efficacy depends on the inoculum size (= how many viruses enter a replication-enabling environment). Also, in [24], we used stochastic simulation to assess prophylactic efficacy after multiple viral challenges. Here, we demonstrate how the proposed method can be used to address these questions.

**Multiple challenges.** Multiple viral challenges can be computed straight-forward as exemplified in the *Methods* section. In Table 2 we show the estimated prophylactic efficacy for different PrEP regimen (either 3 or 7 doses of 2 or 50mg DTG started at $t_0 = 0$) with multiple challenges to a single virus particle, computed using both PGS and EXTRANDE. Foremost, as

**Table 2. Prophylactic efficacy in case of multiple viral challenges.**

| Dose | # Doses | Exposure time | Prophylactic efficacy [%] | | Probability of Infection [%] | |
|------|---------|---------------|---------------------------|-----|------------------------------|-----|
| | | | EXTRANDE | PGS | $P_I(Y_{\{t_i\}}, \mathcal{S})$ | $P_I(Y_{\{t_i\}}, \varnothing)$ |
| 2mg | 3 | 1, 24h | 39.64±1.05 | 40.52 | 11.11 | 18.68 |
| | 3 | 1, 72h | 27.72±1.13 | 27.02 | 13.63 | 18.68 |
| | 3 | 1, 24, 72h | 29.90±0.91 | 29.64 | 18.76 | 26.67 |
| | 7 | 1, 24, 72, 144h | 44.31±0.73 | 44.92 | 18.66 | 33.87 |
| 50mg | 3 | 1, 24h | 82.71±0.59 | 82.58 | 3.25 | 18.68 |
| | 3 | 1, 72h | 72.21±0.74 | 72.27 | 5.18 | 18.68 |
| | 3 | 1, 24, 72h | 73.33±0.60 | 73.79 | 6.99 | 26.67 |
| | 7 | 1, 24, 72, 144h | 87.31±0.37 | 87.25 | 4.31 | 33.87 |

2 or 50mg oral DTG was ingested for 3 and 7 days respectively starting at $t_0 = 0$, and 2–4 viral exposures occurred at the time 'Exposure time' hours after DTG initiation. During each exposure one infectious virus entered a replication-enabling compartment. The corresponding prophylactic efficacy was computed by PGS and EXTRANDE, respectively. For each condition EXTRANDE was run 100 000 times. The 95% confidence bounds of the EXTRANDE estimate was computed using the Greenwoods formula [42]. The probability of infection after multiple viral challenges with- and without drug ($P_I(Y_{\{t_i\}}, \mathcal{S})$ and $P_I(Y_{\{t_i\}}, \varnothing)$) were computed with PGS and are depicted in the last 2 columns. Utilized pharmacokinetic parameters were $k_a = 2.24\text{h}^{-1}$, $V_p/F_{bio} = 0.73$ L, $Q/F_{bio} = 0.0082$ L/h, $CL/F_{bio} = 0.85$ L/h and $V_c/F_{bio} = 17.7$ L.

a sanity check, we can see that both methods yield congruent results for all tested conditions. We can also see that higher dose, as well as a longer time course (seven vs. three days) of DTG dosing improves the prophylactic efficacy, even if more viral challenges occur during the short-course prophylaxis. Also, we observe a interesting interplay between the number of exposures and their timing: For example, if two exposures occur at 1 and 24h after the first dose of DTG vs. three exposures at 1, 24 and 72h after DTG initiation, we see a decrease in efficacy. However, when we compare two exposures occur at 1 and 72h after the first dose of DTG vs. three exposures at 1, 24 and 72h after DTG initiation, we see a slight increase in efficacy. This has the following reason: In the presented example, the prophylactic efficacy (relative risk) after multiple challenges is given by:

$$\varphi(Y_{\{t_i\}}, \mathcal{S}) = 1 - \frac{P_I(Y_{\{t_i\}}, \mathcal{S})}{P_I(Y_{\{t_i\}}, \varnothing)}. \tag{25}$$

where $\{t_i\}$, $i = 1, \ldots, n$ denotes a set of $n$ viral exposures.

Now, if $P_I(Y_{\{t_i\}}, \varnothing)$ increases faster with the number of exposures than $P_I(Y_{\{t_i\}}, \mathcal{S})$, a scenario may arise, in which the prophylactic efficacy (= relative risk reduction) may be higher, although more exposures happened. I.e. the contextual information, '*when did the exposure(s) occur relative to the drug dosing*' is relevant. For example, if many exposures happened at times of almost full protection, the exposed person would be better off than if only a few exposures happened at times of low protection.

**Inoculum size.** Fig 5 shows how the profile of prophylactic efficacy is affected by the number of inoculated viruses. This can be calculated from the solution of the PGS is a straight forward way, akin to Eq (3) (statistical independence assumption). In Fig 5, we observe that different inculum sizes lead to an (exponential) scaling of the prophylactic efficacy, and that

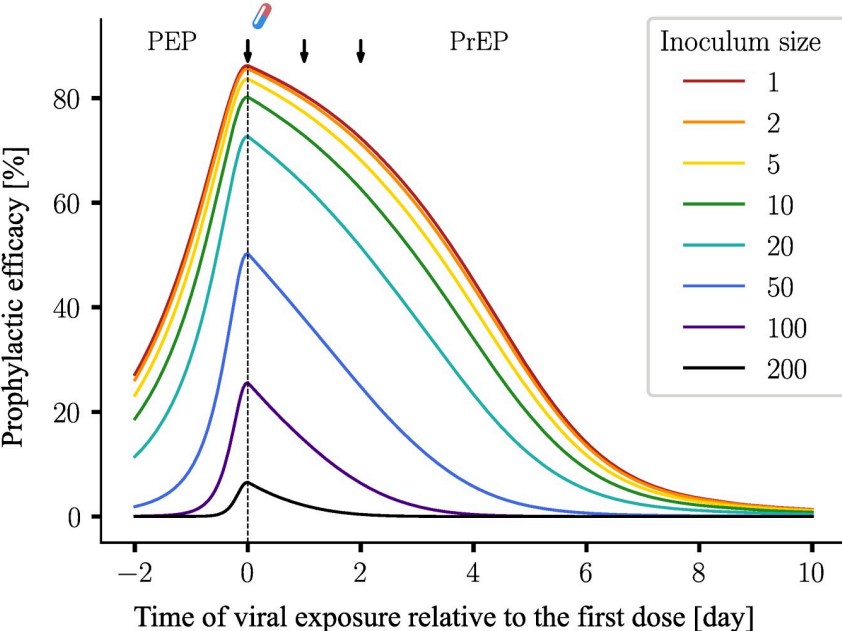

**Fig 5. Profiles of prophylactic efficacy for different inoculum sizes.** The experimental setup was chosen identical to Fig 3. To calculate the prophylactic efficacy profiles, an exponential scaling is applied to the solutions of the PGS (Eq (3)). The solutions are plugged into Eq (1).

the efficacy deteriorates, when large numbers of viruses are able to reach a replication-competent compartment.

## Long-term prophylactic efficacy

Because of its superior computational performance, the PGS can also be applied to estimate prophylactic efficacy over very long time scales for population pharmacokinetics (Pop-PK). I. e., typically pharmacokinetic variability is described by statistical models, such as non-linear mixed effects models (NLME), e.g. in [11]. If the pharmacokinetic characteristics of an individual are not known, a Pop-PK model may still be used to accurately capture likely pharmacokinetic profiles in an individual, given a dosing history. The PGS would then allow to predict the profile of prophylactic protection if the individual was exposed to virus at any time during the observation horizon. Note that this type of analysis is usually not feasible with stochastic methods, due to computational demands (for each time point in the profile, several thousand stochastic simulations would be required).

In Fig 6, we show the estimated long-term efficacy profile for a chronic, 6-month once-daily 50mg oral DTG regimen for different levels of adherence. For each adherence level the computation was conducted on 1000 virtual individuals sampled from the Pop-PK model to

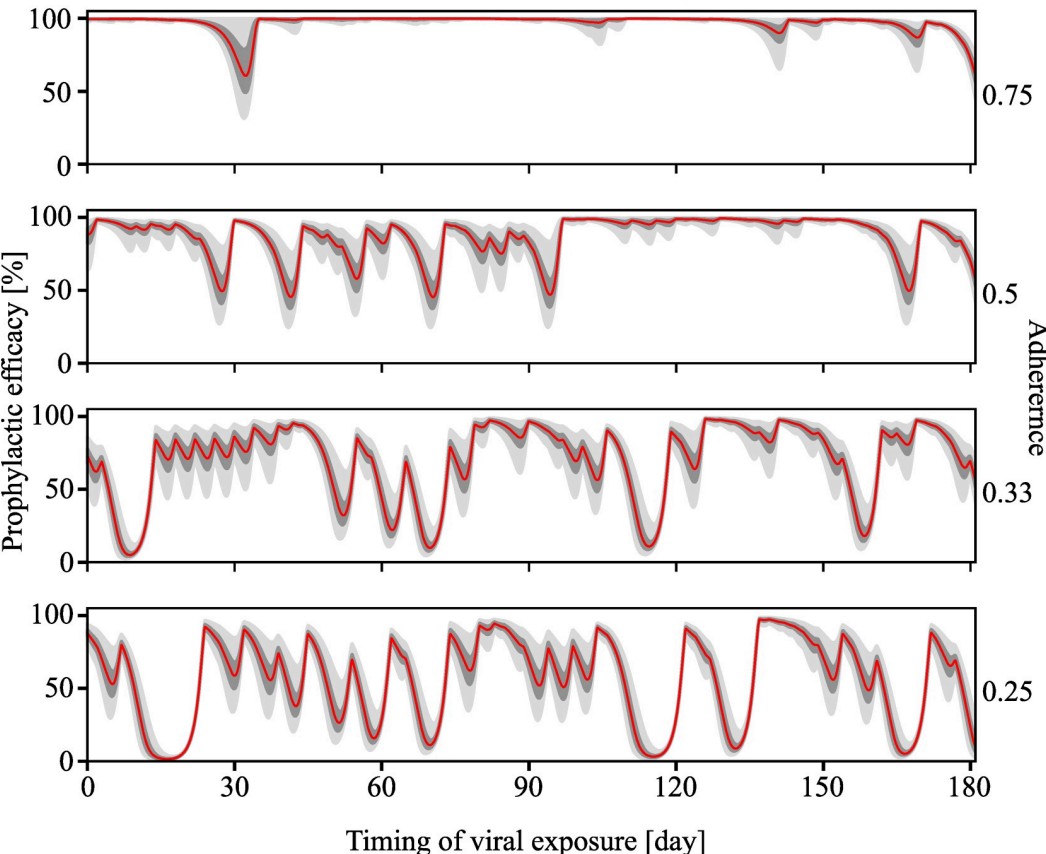

**Fig 6. Long-term prophylactic profile with different levels of adherence.** $N$ = 1000 virtual patients were sampled from the pharmacokinetic parameter distributions defined in Table 2 of [11]. 50 mg dose of DTG / day was ingested in this six-month-long regimen with adherence level of 0.75, 0.5, 0.33 and 0.25. The red line depicts the median predicted prophylactic efficacy, whereas the dark- and light grey areas present the quartile range and the 2.5%–97.5% range respectively. The prophylactic efficacy was computed for $Y_0 = [1, 0, 0]^T$.

capture inter-individual differences in drug pharmacokinetics. The red line and the grey ranges denote the median, interquartile and 2.5%–97.5% ranges of prophylactic efficacy under consideration of inter-individual differences in the pharmacokinetic profiles. With PGS it took about 24 min in total to compute the 6-month prophylactic profiles for 1000 virtual individuals and a given sequence of dosing events (determined by the adherence level), i.e. less than 1.5s for each individual on an AMD R5 core with 3.6Ghz and 16GB RAM (standard laptop). This computation could also be easily parallelized, which would reduce the run time considerably (the entire simulation took about 5 min on the same computer with 12 threads).

## Density function of the extinction event

Using the approaches outlined in the *Methods* section, it is also possible to compute *when* the actual extinction event happens after exposure. This is highly useful in determining how long a prophylaxis on-demand should be given.

Fig 7 shows the cumulative probability of extinction, as well as the density function of the extinction event, computed using the PGS for a 3days 50mg DTG regimen that was initiated at $t_0 = 0$, coinciding with viral exposure. In these simulations, akin to the last example, we sample virtual individuals from the Pop-PK model.

Fig 7A and 7B depict the cumulative-, as well as the density function of the extinction event after exposure to a single initial virus. Fig 7C and 7D show the corresponding distributions after exposure to 20 viruses. From the figures, we can see that the cumulative probability of viral extinction is much lower after exposure to 20- compared to one virus (Fig 7A vs. 7C). Moreover, we can see that, after exposure to a single virus, extinction most likely occurs shortly after exposure, e.g. within 1–2 days. In contrast, when 20 viruses are inoculated, extinction is most likely happening at day 4 after exposure, and that it is still likely that extinction may occur up to 10 days after exposure when a 3days 50mg DTG prophylaxis is applied. Moreover, extinction is less likely to happen: After exposure to 20 viruses, extinction occurs in 10 days with about 75% probability (median), compared to 98% after exposure to a single virus.

In other words, our modelling highlights that the prophylactic efficacy depends on the magnitude of exposure. Moreover, the duration to eliminate the virus is prolonged when more virus becomes inoculated. Essentially, this suggests that large inocula, which may occur after blood transfusions, needle stick exposures, or tissue rupture during sexual contact may require longer duration of prophylaxis to prevent infection.

## Discussion

The prophylactic efficacy of novel drug candidates against HIV is determined by the complex interplay of enzymatic, cellular, viral, immunological, pharmacological, as well as behavioural factors [18]. Some of these factors can be described in experimental surrogate systems [43]. However, their complex interplay, which determines clinical efficacy, can usually not be fully described using *in vitro* or *ex vivo* experiments. Moreover, animal models for HIV prophylaxis are suitable for proof-of-concept studies, but may still be confounded by inter-species differences, as well as differences in the viruses used [44]. Lastly, while clinical studies of HIV prophylaxis are useful to assess the relevant efficacy endpoint, they do not allow to disentangle the complex interplay between the aforementioned factors, because of inherent limitations in the study design, sample sizes and the inability to measure the joint interplay of parameters that determine clinical efficacy. Therefore, it remains a formidable challenge to identify factors that alter prophylactic efficacy, or parameters that can be improved by the development of novel drugs or drug formulations for HIV prophylaxis.

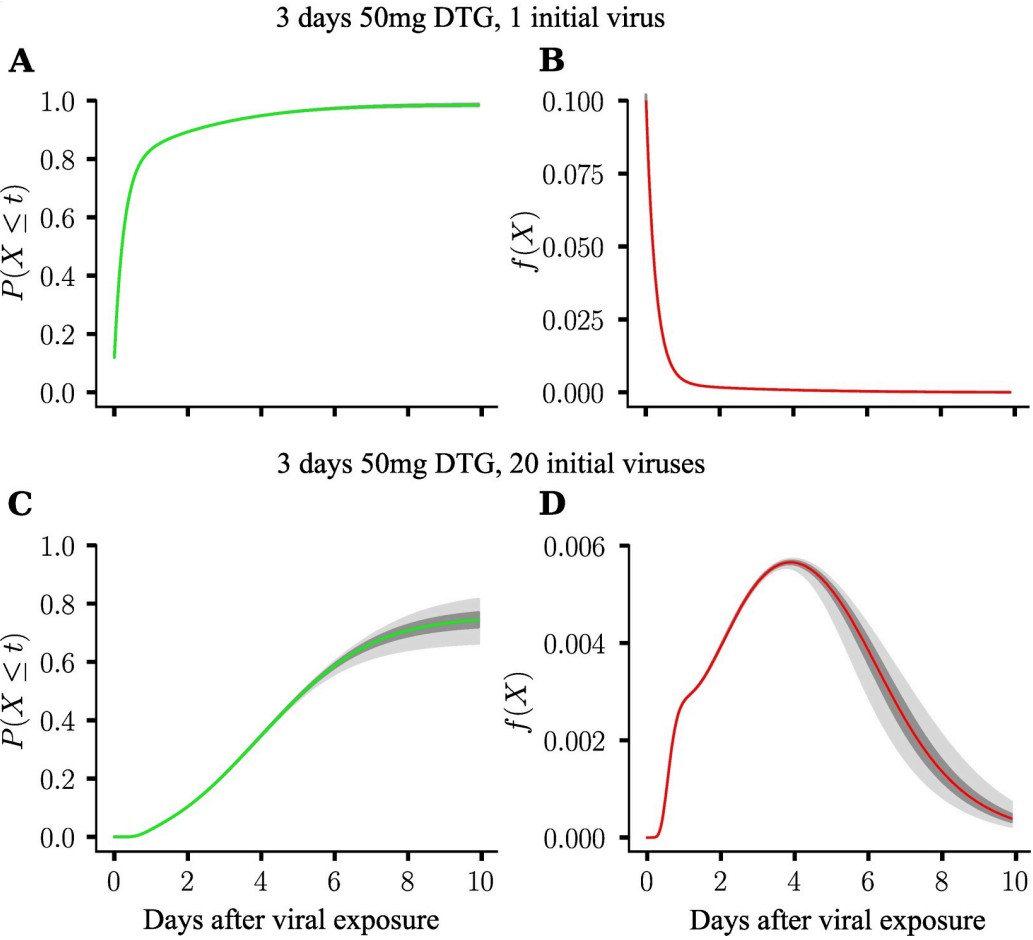

**Fig 7. Cumulative probability and probability density function of extinction event.** $N = 1000$ virtual patients were sampled from the pharmacokinetic parameter distributions defined in Table 2 of [11]. 50 mg dose of DTG / day was ingested for three days, first dose was taken at $t_0 = 0$, coinciding with viral exposure. **A**. Cumulative extinction probability for one initial virus. **B**. Probability density function of extinction for one initial virus. **C**. Cumulative extinction probability for 20 initial viruses. **D**. Probability density function of extinction for 20 initial viruses. The red and green lines depict the respective median values, whereas the dark- and light grey areas present the quartile range and 2.5%–97.5% confidence range, taking inter-individual pharmacokinetic differences into account.

Mathematical models [11, 12, 21–24, 29, 45–47] have become an essential tool to complement our knowledge about prophylactic efficacy of antiviral compounds, as they are able to put the different parameters in context and test their relevance for determining prophylactic efficacy. Stochastic simulation methods are currently the gold standard for estimating prophylactic efficacy from these models [11, 12, 21–24, 29, 46, 47]. Essentially, to estimate prophylactic efficacy with stochastic simulation approaches, a large number of stochastic trajectories is sampled and subsequently classified into infection or extinction events to derive a sample statistic of, for example, the probability of infection $\hat{P}_I(Y_t, \mathcal{S})$. More recently, the extra reaction algorithm for networks in dynamic environments (EXTRANDE) [41] was adapted in [11] to couple pharmacokinetics with intrinsically stochastic viral dynamics following exposure, and to accurately classify stochastic trajectories. Despite the advantages of stochastic approaches, the disadvantages are also very clear: To obtain meaningful statistics, many stochastic simulations need to be conducted to accurately determine the *sample statistics* $\hat{P}_I(Y_t, \mathcal{S})$. The latter

makes the stochastic methods expensive in terms of overall computational time, such that many important *in silico* experiments are infeasible. In particular, the scope of sensitivity analysis with regards to aforementioned factors in integrated, multiscale-models is usually limited. In this work, we introduce a low-dimensional approach to estimate the prophylactic efficacy in considerably less time, in a single run. We envision that the approach can greatly expand the scope of analysis with regards to estimating prophylactic efficacy, by allowing to analyse the long-term effect of prophylaxis, as well as performing sensitivity analysis.

Stochastic simulation methods sample trajectories of the whole system, where any state may arise during simulation. However, in the context of prophylaxis, one is only interested in the probabilities of extinction (and its complement, infection). Therefore, for each state, only those parts that contribute to the extinction event need to be considered. Because the probability of extinction for an arbitrary state can be expressed with the extinction probabilities of the respective unit vectors (Eq (3); statistical independence), only extinction probabilities for unit vectors need to be computed. This is the main idea behind the proposed low-dimensional Ansatz function.

The discrete form of the proposed approach is based on the discrete-time Markov chain of the underlying virus dynamics model. The time was discretized into fixed time steps $\Delta t$ for which the probability flux is computed (Eqs (20)–(22)). In this form, time-dependent functions, e.g. $a_5$, are approximated by a zero-order Taylor approximation for each time step $\Delta t$. The continuous form, Eq (23) can easily be derived from the discrete form by replacing the constant time step $\Delta t$ by an infinitesimally small step $dt$. The method is therefore based on a continuous-time Markov process. In our implementation of the method, we solved the pharmacokinetics beforehand and then wrapped the values of $a_5$ into a function that can be called directly from within the PGS's ODEs, which are solved *backwards in time*. It is notable that a backward ODE solver must be used in the implementation, and the use and configuration of different ODE solvers will have an impact on the accuracy and efficacy of this method.

The set of ordinary differential equations (Eq (23)) derived for the PGS is related to the Kolmogorov backward equations [48], which was used in [22]. The major improvement of our work is to combine this Ansatz with the PK-PD profile of the prophylaxis (Eq (14)) so that the prophylactic efficacy can be computed with arbitrary prophylactic dosing schedules. Notably, instead of-, or in addition to PK-PD profiles, it is also possible to include other time-dependent functions that are relevant to the infection process. For example, any time-dependent, i.e. adaptive immune response could in principle be considered.

In Fig 4, we compared the proposed method with EXTRANDE, which delivers consistent, but less accurate results. Likewise, Table 2 shows the consistency of prediction with PGS and EXTRANDE, regarding the prophylactic efficacy for different DTG regimens with multiple viral exposures.

Using the proposed method, we can also perform analyses that are computationally infeasible with stochastic simulations. As shown in Fig 6, the long-term prophylactic efficacy profiles were computed using PGS with four different values of adherence and a virtual patient cohort of 1000 individuals. This type of analysis allows to quantify sensitivity with regards to adherence and inter-individual pharmacokinetic variability. The analysis showed that DTG can protect highly adherent individuals from acquiring HIV infection and that inter-individual differences are most strongly affecting prophylaxis at times of inconsistent use of the prophylaxis. Moreover, when several consecutive pills are missed, prophylactic efficacy may drop below 50%. Interestingly, the long-term prophylactic efficacy computation took less than 2 seconds for one virtual patient. To compute the corresponding profile with EXTRANDE, for example if an efficacy estimate is to be computed for every hour using 5000 simulations, a total of $24 \times 365 \times 5000 \approx 44$ million simulations for a single adherence level would need to be

conducted. Besides computational time, power consumption (and possible carbon imprint) could therefore be considerably reduced using the proposed methods.

Another possible application of the proposed methods is the possibility to estimate the time of the extinction event, Fig 7. As a showcase of a sensitivity analysis, we estimated, for a 3-days 50mg DTG regimen, the probability density function of the extinction event, when a single, versus 20 viruses were initially reaching a replication-competent physiological compartment. The analysis showed that the time to viral extinction is increased for larger inoculum sizes (more viruses). This analysis thus highlights the complex interplay between viral exposure and prophylaxis that can be analysed with the proposed methods and be used to optimize HIV prophylaxis. With regards to data, interestingly, in vaginal SHIV viral challenge models, which are typically conducted with large inoculum sizes, late viral breakthrough has been observed [49, 50].

The presented method has been derived for the model depicted in Fig 1 and need to be adjusted if other viral dynamics models were used. In S3 Text, we derive the method for an extended viral dynamics model that additionally considers infected T cells, which may turn dormant, e.g. become latently infected. It is well established that these latent reservoirs are a major obstacle to the elimination of HIV during therapy [51]. While these reservoirs are established early in infection, it is unclear whether they alter prophylactic efficacy. To test this hypothesis, we used the proposed methods for the extended viral dynamic model (S3 Text). When comparing the results with those from the simpler model, we however found that the impact of the reservoirs on prophylactic efficacy was negligible.

In summary, we propose a novel method that can estimate the efficacy of arbitrary prophylactic regimen and viral exposures within seconds. The method allows to integrate individual PK/PD profiles and viral dynamics into a single framework and it is more exact than state-of-art hybrid stochastic simulation schemes, like EXTRANDE. We envision that the new method can be applied in many circumstances, in which the stochastic simulation is computationally infeasible, such as parameter sensitivity analysis or long-term efficacy estimation. To this end, the proposed method may even be suitable as part of an App, which may help PrEP users to monitor and plan their PrEP regimen. Moreover, the general schemes may be adapted to study related biomedical questions, like prophylactic efficacy in other pathogens or vaccine efficacy.

## Supporting information

**S1 Text. The supplementary text contains a supplementary derivations for proposed method (distribution of state transition and PGS).**
(PDF)

**S2 Text. The supplementary text contains a complete pseudo-code and the implementation details for the Probability Generating System (PGS).**
(PDF)

**S3 Text. The supplementary text entails the derivation of the equations of PGS for a extended viral dynamic model that contains latently infected viral reservoirs.**
(PDF)

## Author Contributions

**Conceptualization:** Lanxin Zhang, Max von Kleist.

**Formal analysis:** Lanxin Zhang, Junyu Wang, Max von Kleist.

**Funding acquisition:** Max von Kleist.

**Methodology:** Lanxin Zhang, Junyu Wang.

**Project administration:** Max von Kleist.

**Software:** Lanxin Zhang, Junyu Wang.

**Supervision:** Max von Kleist.

**Visualization:** Lanxin Zhang.

**Writing – original draft:** Lanxin Zhang, Max von Kleist.

**Writing – review & editing:** Junyu Wang, Max von Kleist.

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
