## [Decision Letter · Decision Letter 0]

7 Sep 2021

Dear Dr. von Kleist,

Thank you very much for submitting your manuscript "Numerical approaches for the rapid analysis of prophylactic efficacy against HIV with arbitrary drug-dosing schemes" for consideration at PLOS Computational Biology.

As with all papers reviewed by the journal, your manuscript was reviewed by members of the editorial board and by several independent reviewers. In light of the reviews (below this email), we would like to invite the resubmission of a significantly-revised version that takes into account the reviewers' comments.

We cannot make any decision about publication until we have seen the revised manuscript and your response to the reviewers' comments. Your revised manuscript is also likely to be sent to reviewers for further evaluation.

Sincerely,

James R. Faeder

Associate Editor

PLOS Computational Biology

Thomas Leitner

Deputy Editor

PLOS Computational Biology

Reviewer's Responses to Questions

**Comments to the Authors:**

Reviewer #1: A few comments for the authors to consider.

Overall, the model is overly simple for such complicated disease like HIV. The main issue is no latent reservoir is modeled in the primary model. Latent reservoir is critical for HIV eradication, it’s also critical for PrEP. Once the HIV latent reservoir is seeded, there is no way HIV infection can go extinction with current standard-of-care, including DTG. An earlier study showed the latent reservoir seeding can happen as soon as within hours to a few days after initial infection (Whitney et al. Nature 2014 (512):74-77). And many aspects of the HIV infection biology are not yet understood, but it is likely more complicated than the model in this paper.

The authors claimed in the discussions that adding long-lived or latent reservoir to the model seems not impacting the result much. However, the latent reservoir model that the authors described in the Fig S5.1 does not have latency self-renewal and activation mechanism. That would allow HIV goes to extinct in the model, which in reality will never happen. Latently infected cells have great capacity to maintain itself from extinction, and it can become productively infected cell and reentry to the viral replication cycle stochastically, i.e. at least T_L -> 2xT_L and T_L -> T_2 should be modeled in the system. Without these mechanisms, it is not surprising that adding latent reservoir is not impacting the results.

On the other hand, I suspect the approach that authors proposed in the manuscript may have difficulties to handle complex mechanisms such as latent reservoir dynamics more realistically. Immune response is another complex piece in HIV, which is also critical buy omitted in this model.

Reviewer #2: The paper by Zhang et al. explores different approximations to calculate the efficacy of prophylaxis strategies against HIV. The model for prophylaxis is based on viral dynamics and drug pharmacokinetics, and builds on the previous results from this group. Making those methods easily available through approximations is an important step to diffuse these approaches, and the paper is a nice contribution to the field. I have few comments for the authors:

1. I think that the paper is a little long as it is. I would recommend (but worth discussing with the editor) to focus on one approximation in the main document, and to present comparisons with the two others in supp.

2. It’s great that the code is available. If you present your method as a way to empower clinicians/patients, I think it is important to make a shiny application available together with the manuscript? It’s not a huge work but would increase its visibility

3. Even if your study builds on previous works from your lab, it is important that the paper is self-standing. For instance it is difficult to judge the relevance of parameters from Table 1. At least you could show some descriptive statistics for your model predictions in Table 1 (in absence of any tx) : probability of infection according to exposure time (as shown in Table 3), viral kinetic profiles in case of successful infection ?

4. It is unclear to me why you used plasma PK as a driver of efficacy in your model? There is no data in the literature on drug PK in compartments of interest? I would suspect that the drug PK in genital compartments in much less prone to high daily variations than in plasma

5. Sometimes the extinction probability is calculated on V, T1, T2 (Fig 2). But sometimes the efficacy is calculated overall (e.g, Fig 3). It would be helpful if you could clearly define what is your criterion for extinction: one of the compartments, all the compartments? It would be important that you present first the 3 criterions, and then focus on one more specific.

**Have the authors made all data and (if applicable) computational code underlying the findings in their manuscript fully available?**

Reviewer #1: Yes

Reviewer #2: None

PLOS authors have the option to publish the peer review history of their article (what does this mean?). If published, this will include your full peer review and any attached files.

Reviewer #1: No

Reviewer #2: No
---

## [Decision Letter · Decision Letter 1]

3 Dec 2021

Dear Dr. von Kleist,

We are pleased to inform you that your manuscript 'Numerical approaches for the rapid analysis of prophylactic efficacy against HIV with arbitrary drug-dosing schemes' has been provisionally accepted for publication in PLOS Computational Biology.

Best regards,

James R. Faeder

Associate Editor

PLOS Computational Biology

Thomas Leitner

Deputy Editor

PLOS Computational Biology

Reviewer's Responses to Questions

**Comments to the Authors:**

Reviewer #1: I appreciate the authors' effort extending the model to include a more realistic HIV latent reservoir dynamics mechanism. I agree with the authors that the large inoculum size used in the monkey challenge study (Whitney et al. Nature 2014 (512):74-77) may not reflect the real situation in natural human infection. The authors have addressed all my concerns.

Reviewer #2: All my comments have been addressed. I have no further comments

**Have the authors made all data and (if applicable) computational code underlying the findings in their manuscript fully available?**

Reviewer #1: Yes

Reviewer #2: Yes

PLOS authors have the option to publish the peer review history of their article (what does this mean?). If published, this will include your full peer review and any attached files.

Reviewer #1: **Yes: **Youfang Cao

Reviewer #2: No

---

## [Editor Report · Acceptance letter]

17 Dec 2021

PCOMPBIOL-D-21-01254R1 

Numerical approaches for the rapid analysis of prophylactic efficacy against HIV with arbitrary drug-dosing schemes

Dear Dr von Kleist,

I am pleased to inform you that your manuscript has been formally accepted for publication in PLOS Computational Biology. Your manuscript is now with our production department and you will be notified of the publication date in due course.

With kind regards,

Agnes Pap
